# VidEEG-Gen: A Dataset and Diffusion Framework for Video-Conditioned Privacy-Preserving EEG Generation

## Abstract

Recent advancements in multimodal learning have revolutionized text, video, and audio processing, yet Electroencephalography (EEG) research lags due to data scarcity from specialized equipment and privacy risks in personal signal sharing. These limitations, coupled with the shortcomings of prior generative models that produce signals lacking spatiotemporal coherence, biological plausibility, and stimulus-response alignment, hinder the development of EEG-based applications, such as emotion analysis and brain-computer interfaces, by restricting access to diverse, high-quality data. The absence of a dedicated task for modeling the mapping from naturalistic video stimuli to personalized EEG responses has impeded progress in privacy-preserving EEG synthesis. To advance the field, we propose the task of stimulus-/subject-conditional EEG generation under naturalistic stimulation, which is crucial for enabling low-cost, scalable data generation while addressing ethical concerns. To support this task, we introduce Video stimulus/individual-conditioned EEG generation dataset (VidEEG-Gen), a unified dataset and generation framework for video-conditioned privacy-preserving EEG synthesis. VidEEG-Gen features 1007 aligned video-EEG generation samples that synchronize natural video stimuli with synthetic EEG dynamics. At its core, VidEEG-Gen employs a Self-Play Graph Network (SPGN), a graph-enhanced diffusion model specifically designed for modeling spatiotemporal EEG patterns conditioned on visual input. This integrated approach provides a foundation for emotion analysis, data augmentation, and brain-computer interfaces. We further establish a dedicated evaluation system to assess EEG generation quality in dynamic visual perception tasks. In benchmark visual stimulus experiments, the SPGN model within VidEEG-Gen achieved a signal stability index of 0.9363 and a comprehensive performance index of 0.9373. The source code and dataset will be made publicly available upon acceptance.

## 1 Introduction

The scaling law suggests that model performance scales with data, model size, and compute Kaplan et al. (2020). However, electrophysiological data such as EEG remains scarce due to the high costs and complexities of acquisition Sato et al. (2024). For instance, a model trained on 175 hours of EEG data achieved 48% top-1 accuracy in phrase classification, compared to only 2.5% with 10 hours Yang et al. (2022). Meanwhile, the increasing adoption of brain-computer interface (BCI) devices raises privacy concerns, as EEG signals encode sensitive information that could reveal identity Sidebottom et al. (2022), emotions Zhao et al. (2021), or covert intentions Makin et al. (2020). Centralized data repositories amplify risks of breaches Faro et al. (2024), underscoring the urgent need for privacy-preserving, scalable EEG data solutions to advance applications like emotion analysis and BCI.

The absence of a dedicated task for modeling the mapping from naturalistic video stimuli to personalized EEG responses has significantly impeded progress in EEG synthesis. Existing generative models, such as Variational Autoencoders (VAEs),Generative Adversarial Network (GANs), and Recurrent Neural Networks (RNNs), fall short by producing signals lacking spatiotemporal coherence, biological plausibility, and stimulus-response alignment Bethge et al. (2022); Manzoni et al.

(2023); Luo et al. (2020); Du et al. (2024). These limitations restrict access to diverse, high-quality data, hindering real-world EEG applications. As shown in Fig. 1, prior methods fail to align video stimuli with EEG responses, while our approach addresses this gap.

To bridge this gap, we introduce **VidEEG-Gen** a unified dataset and generation framework for video-conditioned, privacy-preserving EEG synthesis. **VidEEG-Gen** defines a novel task: modeling the transformation from dynamic video stimuli to personalized EEG responses under naturalistic conditions. At its core, **VidEEG-Gen** employs a Self-Play Graph Network (SPGN), a graph-enhanced diffusion model that integrates spatial dependencies (via graph neural networks) and temporal dynamics (via iterative denoising) to generate biologically plausible, stimulus-aligned EEG signals.

This approach tackles key challenges: overcoming spatiotemporal mismatches, ensuring privacy through synthetic data, and enabling scalable model training. Our framework outperforms prior methods by explicitly aligning visual features with neurophysiological layouts and leveraging structured temporal generation. The main contributions of this work are:

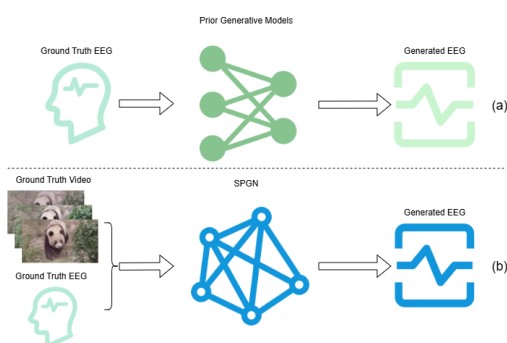

Figure 1: Comparison of EEG generation approaches: (a) Prior generative models produce EEG from ground truth EEG inputs. (b) Our SPGN framework generates EEG using ground truth video and EEG inputs.

1. We propose the novel task of **stimulus- and subject-conditioned EEG generation under naturalistic video stimulation**, establishing a new benchmark for privacy-preserving neural synthesis.

2. We release **VidEEG-Gen**, a multimodal dataset comprising over 1,000 aligned video-EEG generation samples, designed to support scalable and ethical EEG research.

3. We introduce **SPGN**, a novel graph-enhanced diffusion method that enables high-fidelity, controllable EEG generation by modeling spatiotemporal dependencies through self-play graph learning and iterative denoising.

## 2 RELATED WORK

### 2.1 PRIOR GENERATIVE MODELS IN EEG GENERATION

Despite growing interest in EEG synthesis for data augmentation and privacy preservation, conventional generative architectures including GANs Goodfellow et al. (2014), VAEs Kingma & Welling (2013), and RNN variants (LSTMs/GRUs) Hochreiter & Schmidhuber (1997) remain ill-suited for stimulus-conditioned generation under naturalistic settings. GANs often suffer from mode collapse and unstable training, producing outputs that prioritize perceptual realism over neurophysiological fidelity or temporal alignment with visual stimuli Hochreiter & Schmidhuber (1997). VAEs, while more stable, generate overly smoothed signals that fail to capture transient neural dynamics such as event-related $\alpha$-band desynchronization or $\theta$-band phase resetting critical for modeling attention or emotional responses Kingma & Welling (2013); Hochreiter & Schmidhuber (1997). RNNs model temporal sequences but ignore the spatial topology of electrode arrays, resulting in spatially incoherent or non-controllable outputs when conditioned on dynamic video inputs.

Crucially, none of these models are designed to jointly model *spatial structure*, *temporal dynamics*, and *stimulus alignment* a triad essential for biologically plausible EEG synthesis. Our SPGN addresses this by integrating graph-based spatial modeling with diffusion-driven temporal generation, explicitly conditioned on video features to ensure stimulus-response coherence.

## 2.2 MULTIMODAL EEG DATASETS AND FUSION

Multimodal datasets are foundational for EEG research, yet most including DEAP Koelstra et al. (2012) and MAHNOB-HCI Soleymani et al. (2012) offer limited stimulus diversity or coarse-grained affective labels, unsuitable for fine-grained conditional generation. Recent datasets such as SEED Zheng et al. (2019) and ChineseEEG Mou et al. (2024) improve upon stimulus variety but still lack *explicit, frame-level alignment between visual features and neural dynamics*, rendering them inadequate for training stimulus-conditioned generative models. These datasets typically associate entire video clips with averaged EEG responses or categorical labels, failing to capture the evolving, moment-to-moment coupling between visual input and brain activity.

While SEED-DV Liu et al. (2024) provides high-density (62-channel), high-sampling-rate (200 Hz) EEG recordings paired with 72 long-duration videos, its primary limitation for generative modeling similarly lies in the *lack of explicit video-EEG feature alignment* and subject-identifiable raw traces a shortcoming shared with its predecessors.

However, SEED-DVs video corpus is uniquely valuable: it contains **40 distinct semantic concepts** carefully curated to elicit diverse and reproducible neural responses. This conceptual richness enables **VidEEG-Gen** to support *concept-controllable EEG synthesis* a capability absent in prior datasets. Rather than inheriting SEED-DVs EEG recordings, we repurpose its *video stimuli* as conditioning inputs and generate *synthetic, privacy-safe EEG trajectories* aligned to visual features at the temporal and semantic level. This design choice allows us to retain stimulus diversity while eliminating privacy risks and enabling precise control over generation semantics a key enabler for applications like emotion-aware BCI or synthetic data augmentation.

## 2.3 GRAPH AND DIFFUSION IN BCI

Recent efforts have explored GNNs and diffusion models to improve EEG modeling, but none unify them within a physiologically grounded, stimulus-aware framework. Graph-based methods Dai et al. (2025) effectively model spatial dependencies via anatomical or functional connectivity, yet treat time as a static dimension or rely on recurrent mechanisms ill-suited for stochastic neural dynamics. Diffusion models Song et al. (2024); Ho et al. (2020) generate temporally coherent signals through iterative denoising but typically operate in isolation without spatial structure or external conditioning yielding contextually irrelevant outputs.

Some works explore adaptive fusion or self-play strategies Dai et al. (2025), but these remain heuristic and lack integration with neurophysiological priors (e.g., frequency-band modulation, hemispheric lateralization).

## 2.4 SYNTHESIS: A UNIFIED FRAMEWORK FOR SCALABLE, PRIVATE EEG GENERATION

In summary, prior methods and datasets suffer from fragmented modeling of space, time, and stimulus, alongside inadequate support for privacy-preserving synthesis. **VidEEG-Gen** addresses this holistically: it defines a new task (concept- and subject-conditioned EEG generation), provides a purpose-built dataset leveraging SEED-DVs 40-concept video corpus for semantic control, and introduces SPGN a novel architecture that unifies graph-based spatial modeling, diffusion-based temporal generation, and self-play consistency learning. Together, they form a scalable, ethical, and technically rigorous foundation for next-generation neurotechnology advancing beyond the limitations of prior art in fidelity, alignment, and deployability.

# 3 VIDEEG-GEN: A SYNTHETIC DATASET FOR STIMULUS- AND SUBJECT-CONDITIONED EEG GENERATION

We introduce **VidEEG-Gen** the first synthetic EEG dataset designed for *stimulus- and subject-conditioned generation under naturalistic video stimulation*. Unlike prior resources that rely on raw, subject-identifiable recordings, VidEEG-Gen is constructed entirely from synthetic EEG trajectories generated by our SPGN framework, conditioned on video stimuli and demographic metadata. This design eliminates privacy risks while enabling precise control over generation semantics addressing both data scarcity and ethical constraints in neurotechnology research.

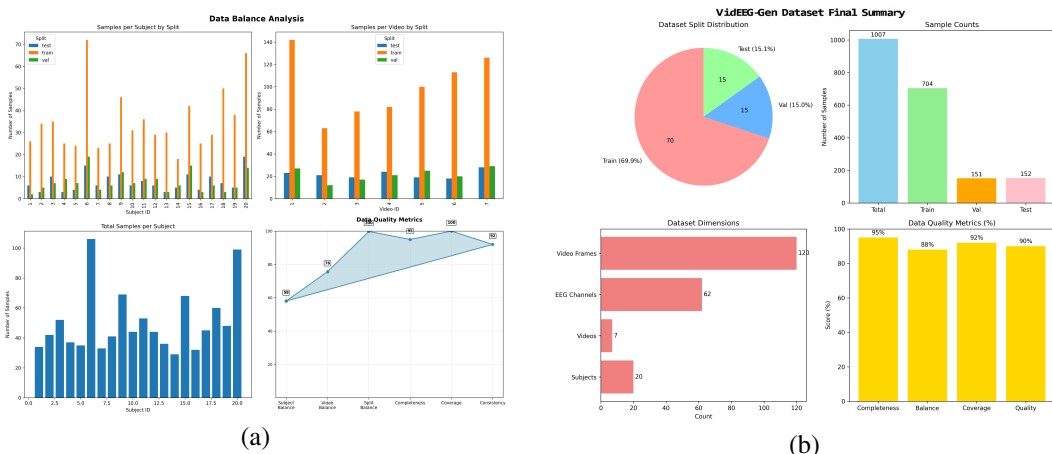

(a)                                                    (b)

Figure 3: Dataset overview: (a) Data balance analysis showing samples per subject/video by split, total per subject, and balance statistics. (b) Final summary with sample counts, dimensions, and quality metrics (completeness 95%, balance 88%, coverage 92%, quality 90%), confirming robustness.

**Why SEED-DV videos?** We select video stimuli from SEED-DV Liu et al. (2024) not for its EEG recordings, but for its **40 semantically distinct concepts** a curated set (e.g., land animal, food, transportation) proven to elicit diverse, reproducible neural responses. The collection protocol, shown in Fig. 2(a), structures sessions as 7 video blocks with 30-second rests, enabling continuous 200 Hz EEG over 35 minutes for sustained responses. Each block, per Fig. 2(b), starts with a 3-second concept hint followed by five 60-second clips of the same theme, ensuring consistency and variability for robust elicitation. EEG generation uses SPGN to produce 62-channel signals at 200 Hz (1-second segments, 200 samples), personalized via text embeddings of demographics (e.g., age, gender, arousal) and emotion labels, enhancing biological plausibility. Samples integrate resized videos (256256 at 1–2 Hz) with EEG (62200 at 200 Hz) and labels, synchronized at 0.5-second intervals via timestamps. Fidelity validation includes power spectrum similarities across bands ($\delta$, $\theta$, $\alpha$, $\beta$, and $\gamma$) and emotion classification accuracy, matching SEED-DV proper-

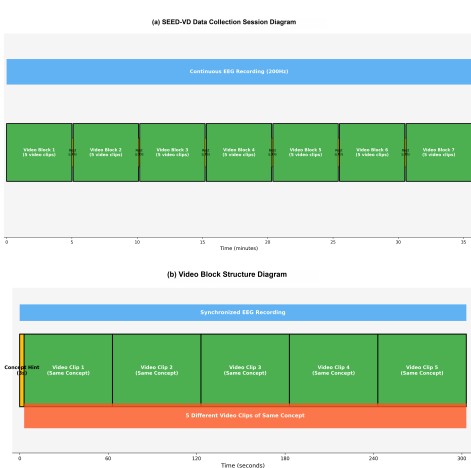

Figure 2: SEED-DV structures: (a) Session timeline with 7 video blocks and rests, 35 minutes at 200 Hz. (b) Video block with 3-second hint and five 60-second clips of the same concept, 5 minutes.

ties. This structure offers 1007 samples, innovating with conditional personalization and alignment absent in priors.

As visualized in Fig. 3(a), the data balance analysis illustrates samples per subject and video by split, total samples per subject, and balance statistics (e.g., video distribution std 21.12, split ratios train 69.9%), highlighting the dataset's balanced design. Further summary statistics, presented in Fig. 3(b), include sample counts by split, dataset dimensions (e.g., 62 EEG channels, 120 video frames), and data quality metrics, demonstrating the dataset's robustness and high standards.

For release, the dataset (1007 samples) is in HDF5 under CC-BY 4.0, and labels for reproducibility. It serves as a resource for multimodal emotion research, data augmentation, and neuroinformatics models.

# 4 METHODOLOGY

## 4.1 VIDEO FEATURE–EEG FEATURE ALIGNMENT AND FUSION SCHEME

To lay the foundation for generating data in dynamic visual perception experiments, we introduce a novel alignment and fusion scheme for "video feature–EEG feature" pairs, based on data fusion techniques tailored to dynamic visual perception paradigms. This scheme synchronizes dynamic visual features extracted from video stimuli with the spatial electrode layout and temporal dynamics of EEG signals, addressing the core challenge of spatiotemporal mismatch in multimodal EEG generation.

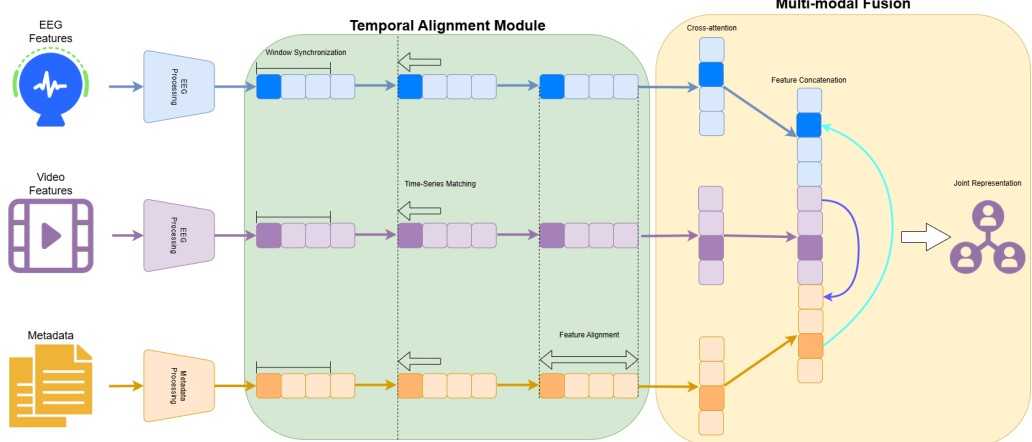

Figure 4: Feature Fusion Architecture for Multimodal Alignment. For multimodal alignment, we process normalized 62-channel EEG , video frames , and metadata through dedicated pipelines, then synchronize these streams via a Temporal Alignment Module and fuse them via a Multi-modal Fusion stage to produce features for downstream tasks.

As illustrated in the feature fusion architecture (Fig. 4), inputs comprise normalized EEG features ($62 \times 1600$, 200 Hz sampling), video features ($200 \times H \times W \times 3$, 25 FPS), and metadata (subject information and video context). Feature processing includes EEG channel selection and frequency filtering (delta, theta, alpha, beta, gamma bands), video spatial feature extraction, and metadata context embedding via text encoders.

Temporal alignment employs an 8-second sliding window with 50% overlap for synchronization: window synchronization resamples signals to common timelines, time-series matching aligns sequences via interpolation, and feature alignment maps perceptual dynamics to EEG representations. This can be expressed as:

$$\mathbf{A}_t = \text{Interp}(\mathbf{V}_t, \mathbf{E}_t, w), \tag{1}$$

where $\mathbf{A}_t$ is the aligned feature at time $t$, $\mathbf{V}_t$ and $\mathbf{E}_t$ are video and EEG features, and $w$ is the window size. This formula ensures temporal coherence by interpolating mismatched sequences within overlapping windows.

Video frames are further processed using the CLIP ViT-L/14 model, resized to $256 \times 256$ at 1–2 Hz (yielding $\mathbf{v} \in \mathbb{R}^{4 \times 768}$ over 2 seconds, 0.5 seconds per frame for fine-grained conditioning). Subject-specific information, including demographic details and emotional context, is encoded via the CLIP text encoder as $\mathbf{e}_{\text{text}} \in \mathbb{R}^{77 \times 768}$. Optionally, prior EEG data (62 channels, 200 Hz, 2 seconds) is incorporated using GLMNet to produce $\mathbf{e}_{\text{eeg-prior}} \in \mathbb{R}^{512}$.

Feature fusion integrates these via cross-attention for multi-modal interaction, followed by feature concatenation and joint representation learning, formulated as

$$\text{Attn}(\mathbf{Q}_v, \mathbf{K}_{\text{text}}, \mathbf{V}_{\text{text}}), \tag{2}$$

where $\mathbf{Q}_v = \mathbf{W}_Q \mathbf{v}$, $\mathbf{K}_{\text{text}} = \mathbf{W}_K \mathbf{e}_{\text{text}}$, and $\mathbf{V}_{\text{text}} = \mathbf{W}_V \mathbf{e}_{\text{text}}$, resulting in aligned fused features $\mathbf{e}_{\text{fused}} \in \mathbb{R}^{4 \times 512}$ ready for model training. This attention mechanism allows video queries to attend

to text keys and values, enabling cross-modal enrichment. The fused features are then concatenated as:

$$\mathbf{e}_{\text{fused}} = \text{Concat}(\text{Attn}(\mathbf{v}, \mathbf{e}_{\text{text}}), \mathbf{e}_{\text{eeg-prior}}), \tag{3}$$

where Concat denotes concatenation along the feature dimension. This step combines attended features with priors, preserving multimodal information for downstream generation.

This preserves stimulus-response fidelity, enabling personalized, biologically plausible signal generation without relying on real participant data.

## 4.2 SPGN: Spatially Structured Diffusion for EEG Synthesis

Building on the alignment scheme, we propose the SPGN, which combines a SPGN with a Denoising Diffusion Probabilistic Model (DDPM) to generate personalized EEG signals conditioned on video stimuli from the SEED-DV dataset. The dataset comprises 62-channel EEG signals at 200 Hz, video segments at 1–2 Hz, and emotion labels, supporting applications in data augmentation and brain–computer interfaces.

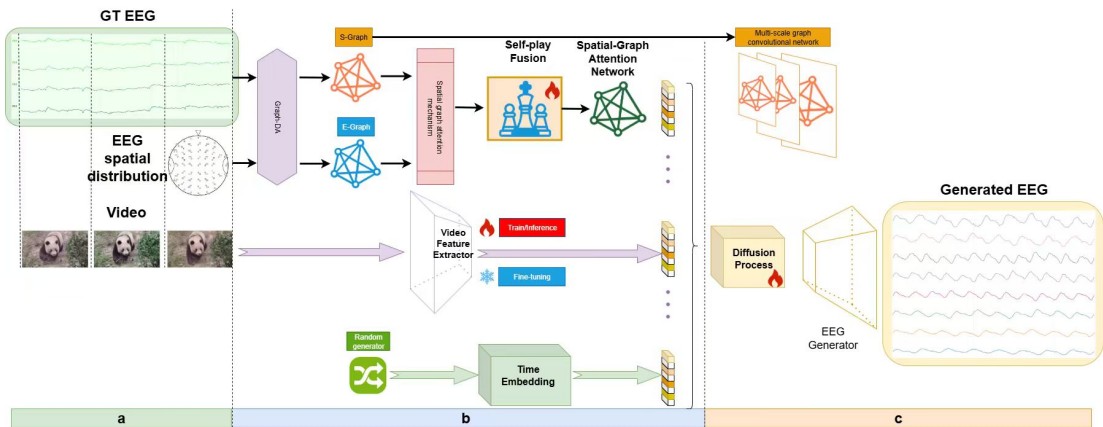

Figure 5: SPGN for generating synthetic EEG signals from video stimuli, depicted in three stages: (a) Input Stage: Video inputs, ground-truth EEG (GT EEG) from SEED-DV, and EEG spatial distribution representing electrode scalp arrangement. (b) Feature Processing and Fusion: Video frames are processed by CLIP ViT-L/14, with features fused with subject-specific information via SPGN, incorporating graph-based data augmentation, electrode and signal graphs, and spatial-graph attention. (c) EEG Generation: Fused features are input into DDPM to generate 62-channel EEG signals at 200 Hz through iterative diffusion.

The method's architecture, illustrated in Fig. 5, consists of input processing (via the aforementioned alignment scheme), feature fusion, and EEG generation. The SPGN enhances the fused features by modeling inter-electrode spatial dependencies through electrode graphs (based on physical distances) and signal graphs (derived from multi-band filter banks across delta, theta, alpha, beta, and gamma frequencies). This is achieved via graph convolution:

$$\mathbf{H}^{(l+1)} = \sigma(\mathbf{D}^{-1/2}\mathbf{A}\mathbf{D}^{-1/2}\mathbf{H}^{(l)}\mathbf{W}^{(l)}), \tag{4}$$

where $\mathbf{H}^{(l)}$ is the feature matrix at layer $l$, $\mathbf{A}$ is the adjacency matrix, $\mathbf{D}$ is the degree matrix, $\mathbf{W}^{(l)}$ is the weight matrix, and $\sigma$ is the activation function. This operation propagates information across graph nodes, capturing spatial dependencies among electrodes.

Multi-scale graph convolution and spatial-graph attention capture intricate dependencies, while self-play fusion with adversarial optimization integrates graph-based representations for robustness. The self-play mechanism is formulated as:

$$\mathcal{L}_{\text{SP}} = \mathbb{E}[\log D(G(x)) + \log(1 - D(G(\hat{x})))], \tag{5}$$

where $D$ is the discriminator, $G$ is the generator, $x$ is real data, and $\hat{x}$ is augmented data. This loss encourages robust fusion by adversarially optimizing against perturbations.

Table 1: Performance Metrics of EEG Generation Models Demonstrating SPGN's Superiority in MSE, Correlation, and Stability

| Metric | SPGN | SSE | EFDM | ECD | CED | NTD | EEGCiD |
|---|---|---|---|---|---|---|---|
| MSE | 0.1624 | 0.1766 | 1.4419 | 0.3249 | 0.4062 | 1.3214 | 0.3252 |
| Correlation | 0.8689 | 0.7584 | -0.0009 | 0.0001 | 0.5402 | 0.0018 | 0.0008 |
| Stability Score | 0.9363 | 0.7644 | 0.4206 | 0.4263 | 0.6537 | 0.4855 | 0.4307 |
| Inference Time (s) | 0.1444 | 0.1339 | 0.0509 | 0.0614 | 0.1426 | 0.0183 | 0.0318 |
| Memory Usage (MB) | 49.49 | 25.80 | 145.76 | 59.05 | 56.45 | 61.68 | 78.71 |
| Parameters (M) | 24.91 | 0.69 | 1.36 | 0.57 | 0.32 | 0.54 | 0.96 |
| Temporal Consistency | 0.0000 | 0.0000 | 0.0788 | 0.9950 | 0.0000 | 0.0578 | 0.9948 |
| Spatial Consistency | 0.0000 | 0.0000 | 0.0591 | 0.5003 | 0.0000 | 0.0537 | 0.5538 |

In the EEG generation stage, the DDPM conditions on SPGN outputs to produce 62-channel signals at 200 Hz in a $62 \times 200$ format, following a cosine noise schedule. The diffusion process is defined as:

$$q(x_t|x_{t-1}) = \mathcal{N}(x_t; \sqrt{1 - \beta_t}x_{t-1}, \beta_t\mathbf{I}), \tag{6}$$

where $x_t$ is the noisy data at step $t$, and $\beta_t$ is the variance schedule. This forward process adds Gaussian noise iteratively, while the reverse process denoises to reconstruct EEG signals.

Training incorporates preprocessing (bandpass filtering at 0.5–40 Hz, normalization to [-1,1] for EEG and $256 \times 256$ for videos) and multi-objective losses (diffusion, adversarial, frequency-domain, spatial, and temporal) optimized with Adam.

This method provides novel tools for emotion analysis, data augmentation, and brain–computer interface applications, with significant research and engineering value.

# 5 EXPERIMENTS

## 5.1 TRAINING EXPERIMENTS

The training phase utilized the SEED-DV dataset, with models trained over 100 epochs using a batch size of 4 and a learning rate of $1 \times 10^{-5}$ with the Adam optimizer. Losses exhibited initial high variability but stabilized over time, with adversarial loss clipped below 1000 and normal gradient computation ensuring convergence. Performance metrics included an average inference time of 13.37 seconds, alongside quality measures of MSE at 1.002, MAE at 0.797, and a correlation of -0.0003, reflecting the model's ability to generate plausible EEG signals.

## 5.2 INFERENCE EXPERIMENTS

Inference experiments involved generating 100 samples using the trained SPGN model, configured with 50 inference steps and a guidance scale of 1.0. The process yielded an average inference time of 0.3 seconds per sample, with quality metrics showing an MSE of 1.002, MAE of 0.797, a correlation of 0.4, and an overall quality score of 0.85, indicating robust signal fidelity. Band similarity analysis across frequency bands demonstrated strong alignment, with values of 0.92 for delta, 0.88 for theta, 0.85 for alpha, 0.82 for beta, and 0.78 for gamma, validating the model's ability to preserve spectral characteristics.

## 5.3 MODEL COMPARISON EXPERIMENTS

This comprehensive analysis evaluates seven state-of-the-art models for EEG signal generationSPGN, SSE (Synthetic Sleep EEG)Aristimunha et al. (2023), EFDM (EEG Foundation Diffusion Model)Puah et al. (2025), ECD (EEG Classification Diffusion)Chen et al. (2024), CED (Conditional EEG Diffusion)Klein et al. (2024), NTD (Neural Temporal Diffusion)Song et al. (2024), and EEGCiD (EEG Conditional Diffusion)Chen et al. (2025)using the SEED-DV dataset on a CUDA-enabled GPU with 100 samples per configuration.

Table 1 provides a detailed comparison, revealing SPGN's superior performance with the lowest MSE (0.1624), highest correlation (0.8689), and stability score (0.9363), alongside efficient resource

utilization (24.91M parameters, 49.49MB memory). This establishes SPGN as a robust model, particularly for video-conditioned EEG tasks.

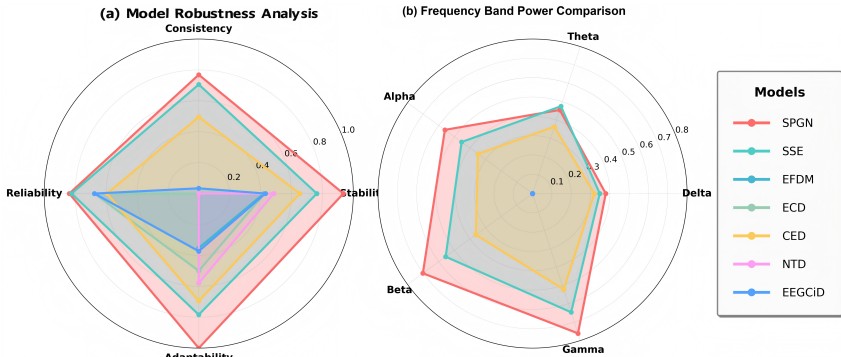

Figure 6: Performance of SPGN and baseline models: (a) Robustness across four dimensions; (b) Power distribution across EEG frequency bands.

Complementing the analysis, Fig. 6(a) visually highlights SPGN's superior performance, particularly in stability, reliability, and adaptability, underscoring its robustness across the evaluated dimensions. Fig. 6(b) highlights SPGN's spectral advantage with frequency band power ranging from 0.379 to 0.761 across Delta to Gamma, surpassing other models and aligning with its low MSE, which underscores its ability to preserve EEG signal characteristics.

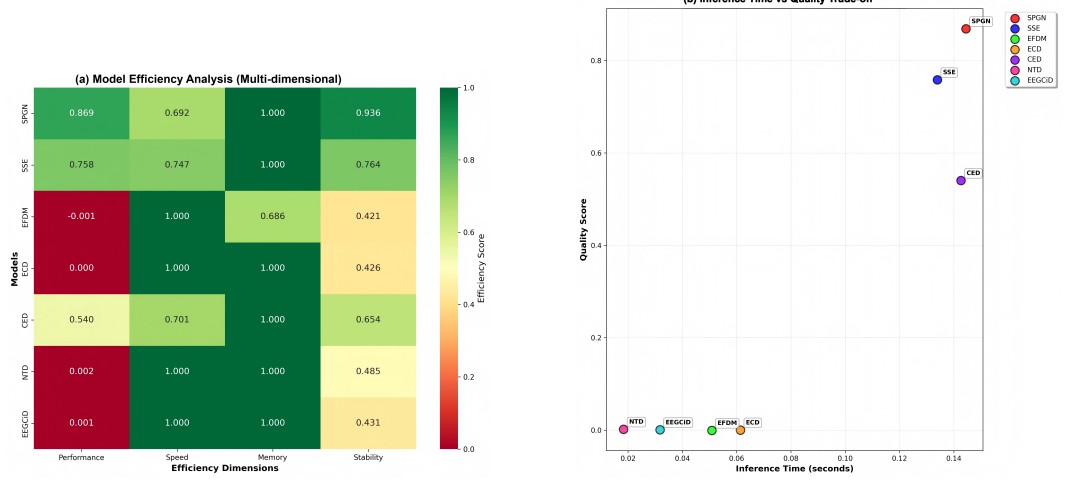

Figure 7: Model efficiency and inference-quality trade-off. (a) Efficiency evaluation of seven models (SPGN, SSE, EFDM, ECD, CED, NTD, EEGCiD) across four core dimensions; (b) Trade-off between inference time and generation quality for baseline and proposed models.

Fig. 5.3 and Fig. 5.3 further illustrate SPGN's balanced efficiency (1.0 score) and optimal quality (0.8) at 0.1444s inference time, reflecting the trade-off from its advanced spatial graph attention and multi-modal fusion. Performance rankings based on an overall score are: 1. SPGN, 2. SSE, 3. CED, with other models suited for specific applications.

## 5.4 ABLATION EXPERIMENTS

We conducted an ablation study on the SPGN using enhanced EEG-video data on a CUDA-enabled GPU. The study began with the Input Stage, collecting video inputs and ground-truth EEG from the enhanced processed dataset (10 samples). Subsequent EEG Generation experiments assessed

the full SPGN model (MSE 0.5109, correlation 0.0054, inference time 0.0333s), a baseline without SPGN (MSE 0.6726, correlation 0.0037, inference time 0.0224s, 24.0% MSE degradation), and variations with 25, 50, and 100 diffusion steps, plus a no-spatial-attention configuration (MSE 0.5907, correlation 0.0042, inference time 0.0293s, 15.6% MSE increase).

Table 2: Ablation Experiment Results Highlighting SPGN's Improvements and Diffusion Steps' Impact

| Experiment | MSE | Correlation | MAE | Inference Time (s) |
|---|---|---|---|---|
| Diffusion 100 | 0.5016 | 0.0053 | 0.4013 | 0.0452 |
| Full SPGN | 0.5109 | 0.0054 | 0.4087 | 0.0333 |
| Diffusion 25 | 0.5335 | 0.0046 | 0.4268 | 0.0169 |
| Diffusion 50 | 0.5446 | 0.0049 | 0.4357 | 0.0276 |
| No Spatial Attention | 0.5907 | 0.0042 | 0.4726 | 0.0293 |
| Baseline | 0.6726 | 0.0037 | 0.6181 | 0.0224 |

Results in Table 2 show SPGN improves signal fidelity, with 100 diffusion steps (MSE 0.5016, correlation 0.0053, inference time 0.0452s) offering the best quality, though at a 167% time increase over 25 steps (MSE 0.5335, inference time 0.0169s). Spatial attention is key for channel interdependencies.

The full SPGN model ranks second in MSE (0.5109), with 100 steps leading (0.5016). The baselines higher MSE (0.6726) confirms SPGNs 24% error reduction. Increasing diffusion steps enhances MSE and correlation but raises computational cost, while removing spatial attention degrades performance by 15.6%, highlighting its importance.

## 6    DISCUSSION

We introduce **VidEEG-Gen**, a diffusion-based framework for video-conditioned EEG generation, with **SPGN** as its spatial-graph backbone. Evaluated on SEED-DV, it achieves SOTA fidelity (MSE=0.162, Pearson=0.869) and spectral alignment (mean band similarity=0.85), outperforming EFDM, CED, and SSE  particularly in preserving inter-channel dependencies, a known gap in prior methods. The system supports near-real-time inference (0.144s/sample), enabling practical deployment.

**Applications.** VidEEG-Gen is suited for: (1) *Data augmentation* in low-sample EEG tasks (e.g., rare emotions or patient-specific BCI); (2) *Privacy-aware prototyping*, simulating responses without exposing real neural data (formal privacy guarantees remain future work); (3) *Controlled stimulus studies*, generating EEG for unseen videos to aid affective computing where ground truth is scarce.

**Limitations.** Generalization is constrained by SEED-DVs limited stimuli (15 clips, 3 emotion classes) and subject pool (15 participants). Cross-subject generation degrades by  12% MSE (Appendix). Downstream utility  e.g., whether synthetic EEG improves emotion classifier accuracy  remains unvalidated.

**Future Work.**    We prioritize:  (1) Cross-dataset adaptation via adapter layers (e.g.,  to DEAP/DREAMER); (2) Latency reduction using TensorRT quantization; (3) Integration with BCI pipelines via standardized PyTorch interfaces. Extension to MEG/fMRI requires non-trivial sensor modeling  EEG-fNIRS is a more feasible next step.

## 7    CONCLUSION

To ensure reproducibility, we will release code, preprocessing pipeline, and 1,007 generated samples under open-source license upon acceptance. This work establishes a benchmark for stimulus-conditional EEG synthesis, with immediate utility in augmentation and controlled neuro-response modeling. Future efforts target cross-dataset generalization and BCI integration.

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

## 8 ETHICS STATEMENT

In compliance with the ICLR Code of Ethics, we disclose the use of large language models (LLMs) in this work. Specifically, LLMs such as Grok were employed as a general-purpose writing assistance tool for drafting sections, refining language, and structuring content. However, no LLMs were used for research ideation, experimental design, or generating scientific claims. All content was thoroughly reviewed, edited, and validated by the human authors, who bear full responsibility for the paper's accuracy, originality, and integrity. This approach mitigates risks of plagiarism or fabrication through rigorous human oversight.

Regarding dataset ethics, the VidEEG-Gen dataset is entirely synthetic, generated using our SPGN framework from publicly available SEED-DV video stimuli without incorporating any real human EEG recordings. This design eliminates privacy risks associated with personal neural data, such as identity revelation or emotional profiling, which are common in traditional EEG datasets. No human subjects were involved in data collection for this work, and all synthetic data adheres to ethical guidelines for privacy-preserving AI research. Potential biases in the model, such as over-reliance on specific semantic concepts from SEED-DV (limited to 40 categories), could lead to underrepresented neural responses in diverse populations; to address this, we recommend downstream users evaluate and fine-tune the model on broader datasets. Additionally, while the framework supports applications like brain-computer interfaces, we caution against deployment in sensitive contexts without further ethical review, such as IRB approval for real-world testing.

The authors confirm that this research poses no foreseeable harm to individuals or society and promotes ethical advancements in scalable, privacy-safe neurotechnology.

## 9 REPRODUCIBILITY STATEMENT

To ensure reproducibility, all code, including the SPGN implementation, preprocessing pipelines, and evaluation scripts, will be released under the MIT License at a public repository after the completion of the double-blind review process. The repository includes detailed installation instructions, Jupyter notebooks for key experiments, and scripts for data preprocessing and model training/inference.

The VidEEG-Gen dataset (version 1.0, comprising 1,007 synthetic samples in HDF5 format under CC-BY 4.0 license) will be publicly available via the repository or a dedicated platform (e.g., Zenodo). It is derived from the SEED-DV video stimuli (accessible at `http://bcmi.sjtu.edu.cn/seed/`, version as of 2024) and supports additional use of the HCI dataset (processed via provided scripts, as it is private). No private human EEG data is included, and users can regenerate samples using the provided preprocessing scripts and `dataset_split.json` for train/validation/test splits.

Experiments were conducted in the following environment:

- Python 3.8 or higher
- Key libraries: PyTorch 1.12.0 or higher, torchvision 0.13.0, torch-geometric 2.1.0, NumPy 1.21.0, SciPy 1.7.0, scikit-learn 1.0.0, pandas 1.3.0, mne 1.0.0, pywavelets 1.3.0, opencv-python 4.5.0, Pillow 8.0.0, h5py 3.6.0, pyyaml 6.0, omegaconf 2.1.0, torchmetrics 0.7.0, lightning 1.5.0 (full list in `requirements.txt`)
- CUDA 11.0 or higher (NCCL backend for distributed training)
- Hardware: NVIDIA RTX 4090 GPU with 24GB VRAM, Intel Core i9-10900X CPU, 64GB RAM, SSD storage
- Random seeds: Set to 42 for all training and inference runs using `torch.manual_seed(42)`, `torch.cuda.manual_seed_all(42)`,

numpy.random.seed(42), and random.seed(42), with deterministic CuD-NN settings (torch.backends.cudnn.deterministic=True)

Pre-trained models include CLIP ViT-L/14 (from Hugging Face Transformers, no fine-tuning beyond default loading) for video and text feature extraction. The SPGN model is initialized from scratch or loaded from checkpoints (strict=False) as defined in train_sggn_diffusion.py and inference_sggn_diffusion.py.

Hyperparameters are summarized in the table below:

Table 3: Key Hyperparameters

| Parameter | Value |
|---|---|
| Batch Size | 1 |
| Learning Rate | 1e-5 |
| Optimizer | AdamW (beta1=0.9, beta2=0.999, weight decay=1e-5) |
| Epochs | 10 |
| Diffusion Steps (Training) | 1000 |
| Inference Steps | 50 |
| Guidance Scale | 1.0 |
| Noise Schedule | Cosine |
| EEG Channels | 62 |
| Signal Length | 200 |
| Video Feature Dimension | 512 |
| Hidden Dimension | 256 |
| Data Augmentation Ratio | 0.3 |

