# OpenReview forum: "VidEEG-Gen: A Dataset and Diffusion Framework for Video-Conditioned Privacy-Preserving EEG Generation"
_ICLR.cc/2026/Conference — ICLR 2026 Conference Withdrawn Submission_

### Official Review · Reviewer_Ga3Y · 2025-10-20

**Soundness:** 2
**Presentation:** 2
**Contribution:** 2
**Rating:** 2
**Confidence:** 4

**Summary:**

This paper introduces VidEEG-Gen, a framework addressing the scarcity and privacy concerns associated with EEG data. The authors propose a new task: generating personalized, synthetic EEG signals conditioned on naturalistic video stimuli. To support this, they present a synthetic dataset derived from the SEED-DV video stimuli and a novel generative model called Self-Play Graph Network (SPGN). SPGN is described as a graph-enhanced diffusion model designed to capture the spatiotemporal dependencies in EEG signals while being conditioned on video features and subject metadata (e.g., demographics). The goal is to produce biologically plausible, stimulus-aligned EEG data that preserves privacy by avoiding the use of real subject recordings. The authors evaluate SPGN against several other EEG generation models, claiming superior performance in signal fidelity, stability, and spectral characteristics.

**Strengths:**

The paper tackles the critical issues of data scarcity and privacy in EEG research, which are significant barriers in the field.
Proposing the specific task of video-conditioned, personalized EEG generation is potentially valuable for advancing stimulus-response modeling.
The SPGN model attempts to explicitly model both spatial (graph) and temporal (diffusion) aspects of EEG, which is methodologically relevant.

**Weaknesses:**

The core contribution, the VidEEG-Gen dataset, is generated by the proposed model (SPGN) itself. The entire evaluation framework appears to operate within this synthetic domain, lacking grounding in real EEG data distributions.
There is no evidence presented showing that the synthetic EEG generated by SPGN accurately reflects the characteristics (dynamics, spectral properties, spatial patterns) of real EEG recorded in response to the SEED-DV videos. Claims of "biological plausibility" are entirely unsubstantiated.
The paper does not clearly explain how the initial "ground truth" synthetic EEG signals (used for training SPGN and evaluating all models) were created or validated.

**Questions:**

Please clarify precisely how the "ground truth" synthetic EEG signals used for training the SPGN model and for evaluating all methods in Table 1 were generated. What ensures their fidelity to real EEG responses elicited by the SEED-DV videos?

Why was the evaluation not performed by training models on real SEED-DV EEG data (or another suitable real dataset) and evaluating their ability to generate plausible signals conditioned on video, perhaps assessing quality via downstream tasks or established distribution metrics (like FID adapted for EEG)?

What specific mechanism allows SPGN to generate personalized EEG based on metadata? How was this personalization capability validated?

What is the definition and validation for the "signal stability index" and "comprehensive performance index" metrics used to claim SOTA performance?

---

### Official Review · Reviewer_gfqH · 2025-10-26

**Soundness:** 2
**Presentation:** 2
**Contribution:** 2
**Rating:** 4
**Confidence:** 3

**Summary:**

The paper defines a new task of stimulus- and subject-conditioned EEG generation under naturalistic video stimulation. The paper also introduces VidEEG-Gen, a unified dataset and framework to study it. VidEEG-Gen contains 1,007 samples that align video clips (drawn from the SEED-DV corpus for semantic diversity) with synthetic EEG trajectories. The method centres on SPGN, a graph-enhanced diffusion model that fuses video features, subject metadata and optional EEG priors via a dedicated alignment and fusion pipeline. The approach then models inter-electrode dependencies with electrode and signal graphs while using diffusion for temporally coherent synthesis. The paper also establishes an evaluation protocol (including spectral band similarity, correlation, stability and composite scores) and reports that SPGN outperforms several recent EEG generative baselines.

**Strengths:**

- The paper introduces the task of stimulus- and subject-conditioned EEG generation under naturalistic video stimulation, whic is an interesting contribution. This redefines EEG synthesis as a multimodal mapping from visual input to biologically plausible neural dynamics, rather than data-driven signal reconstruction. The accompanying VidEEG-Gen dataset establishes the first benchmark for this setting.

- The SPGN architecture represents a synthesis of graph neural networks (for electrode-level spatial structure) and diffusion models (for temporal consistency). This integration addresses the limitations of earlier GAN- or VAE-based EEG generators due to the lack of spatiotemporal coherence and stimulus-response alignment.

- The paper presents the architecture and preprocessing pipeline in a structured and detailed way, including the temporal alignment, multimodal fusion and spatial graph construction processes. It is clear how EEG, video and text features interact across modules, to make reproducibility and easy understanding possible.

**Weaknesses:**

- While the proposed task and dataset are conceptually novel, the work’s significance remains constrained by the narrow empirical scope. All experiments are based on video stimuli from SEED-DV, which covers only 15 participants and 40 concepts. The authors acknowledge that cross-subject generalisation degrades by 12% in mean squared error and that downstream utility (e.g., whether the generated EEG improves classifier performance) is untested.

- The same SPGN model both defines and evaluates the dataset’s structure. The biological plausibility and realism of the synthetic signals are assessed through internal metrics (e.g., frequency-band similarity, stability index) but not through expert or empirical comparison with real EEG traces. This limits confidence in the dataset’s physiological credibility.

- The paper presents multiple interacting modules, including CLIP-based video encoders, text embeddings, graph convolutions, adversarial self-play and diffusion steps, but the ablation study explores only a small subset of these components (spatial attention and diffusion step count). As a result, it is unclear which design elements contribute most to the observed improvements.

**Questions:**

- The paper mentions that the EEG signals in VidEEG-Gen are generated entirely by the SPGN model and not derived from real EEG recordings. Could the authors clarify how they prevent circularity between dataset construction and model evaluation? Specifically, is the same trained SPGN model used both to create and to benchmark the dataset?

- In Section 4.1, the paper describes a fusion pipeline that aligns CLIP-extracted video features, demographic text embeddings and optional EEG priors using cross-attention. Could the authors clarify the role of the EEG prior in this process? For instance, is the prior always available during training, or is it optional to simulate unseen-subject conditions?

---

### Official Review · Reviewer_cDo3 · 2025-10-30

**Soundness:** 2
**Presentation:** 3
**Contribution:** 1
**Rating:** 0
**Confidence:** 4

**Summary:**

The paper presents a dataset for privacy-preserving EEG synthesis.
Synthetic data generation may be important for increasing augmented data available for applications, but I do not understand why we need a dataset for this task. Normally, we’d like to augment data from a particular recording setup and task.

**Strengths:**

- None

**Weaknesses:**

-There is no description of data or recording from subjects
- This is not dataset contrbution
- Concept controllable EEG data is not considered to be valid. There is some evidence of neural correlates that differ across semantic stimuli, but generally this mapping does not exist as EEG is very noisy and encodes attention, but not semantics.
- The paper claims to generate diverese synthetic responses, but I think this is overclaiming
- If the approach is syntethic data generation, then why is the paper written as dataset contribution?

**Questions:**

See weaknesses

---

### Official Review · Reviewer_iUcf · 2025-11-05

**Soundness:** 3
**Presentation:** 3
**Contribution:** 3
**Rating:** 4
**Confidence:** 3

**Summary:**

This paper introduces VidEEG-Gen, a dataset and framework for video-conditioned, privacy-preserving EEG generation. The authors define a new task—stimulus- and subject-conditioned EEG synthesis using naturalistic video inputs—and propose the Self-Play Graph Network (SPGN), a graph-based diffusion model designed to more faithfully capture the spatial, temporal, and semantic relationships in EEG. The work includes the release of a 1007-sample synthetic EEG dataset aligned to video stimuli, reference implementations, and comparative/ablation studies benchmarking the approach against recent alternatives.

**Strengths:**

The paper tackles the important issue of data scarcity and privacy in EEG research, proposing synthetic generation as a practical solution for applications in brain-computer interfaces and emotion analysis.
The SPGN framework judiciously combines graph neural networks (for capturing inter-electrode spatial dependencies) with denoising diffusion probabilistic models and cross-modal alignment/fusion mechanisms.
The work includes quantitative comparisons across multiple recent generative baselines, highlighting SPGN's solid performance in terms of signal fidelity and computational efficiency.

**Weaknesses:**

Explorations of cross-modal conditioning or alternative fusion architectures are not exhaustively presented—raising questions about generality.
The proposed dataset, while meticulously constructed, is generated using only SEED-DV video stimuli and is of modest size.
The generalizability of the approach to more diverse or non-Chinese video/subject populations, or other brain recording scenarios, is acknowledged as a limitation, but no quantitative cross-dataset evidence is provided.

**Questions:**

Could you detail how "spatial-graph attention" in the SPGN is formulated and how it interacts with the denoising diffusion process at each step? Is it applied as a preprocessing step, or jointly with diffusion iterations?
What is the practical impact of using solely synthetic EEG for both training and evaluation? Are there risks of model feedback loops or degraded transfer performance to real data scenarios?

---

### Note · Authors · 2025-11-12

**Comment:**

Dear ICLR 2026 Program Chairs and OpenReview Team,

We, the authors of submission #22779 titled “VidEEG-Gen: A Dataset and Diffusion Framework for Video-Conditioned Privacy-Preserving EEG Generation”, hereby formally request to withdraw our manuscript from consideration for ICLR 2026.

This decision has been made unanimously by all co-authors after carefully reviewing the feedback provided by the reviewers during the review process. We appreciate the time and thoughtful comments from the reviewers and acknowledge the concerns raised regarding the synthetic nature of the dataset, validation against real EEG data, and methodological clarity. While we believe the proposed task and framework hold potential, we agree that substantial revisions and additional empirical validation are necessary before resubmission.

In accordance with the OpenReview withdrawal policy, we confirm that:

All authors consent to this withdrawal.
The withdrawal is voluntary and not due to a double-submission violation or ethical breach.
We understand that once withdrawn, the submission will no longer be considered for ICLR 2026, and its status will be marked as “Withdrawn” on OpenReview.
Thank you for your support and for facilitating a transparent and constructive review process.

Sincerely,

Yunfei Guo, Tao Zhang, Wu Huang, Yao Song

On behalf of all authors

**Withdrawal Confirmation:**

I have read and agree with the venue's withdrawal policy on behalf of myself and my co-authors.